# Phospholipases C and D and Their Role in Biotic and Abiotic Stresses

**DOI:** 10.3390/plants10050921

**Published:** 2021-05-04

**Authors:** Víctor M. González-Mendoza, M. E. Sánchez-Sandoval, Lizbeth A. Castro-Concha, S. M. Teresa Hernández-Sotomayor

**Affiliations:** 1CONACyT- Centro de Investigación en Alimentación y Desarrollo, A.C. Unidad Regional Hidalgo, San Agustín Tlaxiaca, Hidalgo 42163, Mexico; victor.gonzalez@ciad.mx; 2Department of Medical Biochemistry and Cell Biology, University of Gothenburg, 41390 Gothenburg, Sweden; eugensanch@gmail.com; 3Unidad de Bioquímica y Biología Molecular de Plantas, Centro de Investigación Científica de Yucatán, Mérida, Yucatán 97205, Mexico; liarcaco@cicy.mx

**Keywords:** phospholipases, plant immune response, biotic stress

## Abstract

Plants, as sessile organisms, have adapted a fine sensing system to monitor environmental changes, therefore allowing the regulation of their responses. As the interaction between plants and environmental changes begins at the surface, these changes are detected by components in the plasma membrane, where a molecule receptor generates a lipid signaling cascade via enzymes, such as phospholipases (PLs). Phospholipids are the key structural components of plasma membranes and signaling cascades. They exist in a wide range of species and in different proportions, with conversion processes that involve hydrophilic enzymes, such as phospholipase-C (PLC), phospholipase-D (PLD), and phospholipase-A (PLA). Hence, it is suggested that PLC and PLD are highly conserved, compared to their homologous genes, and have formed clusters during their adaptive history. Additionally, they generate responses to different functions in accordance with their protein structure, which should be reflected in specific signal transduction responses to environmental stress conditions, including innate immune responses. This review summarizes the phospholipid systems associated with signaling pathways and the innate immune response.

## 1. Introduction

All stimuli start at the surface and are detected by components of the plasma membrane, where a molecular receptor initiates lipid cascade signaling [1,2]. This can manifest as early or late signaling events that occur within minutes or hours after environmental responses and determine the outcome of stress triggers [3]. Phospholipid signaling responses have developed interrelated communication with immune systems to protect against the invasion of pathogenic microorganisms [4,5]. One system of phospholipid signal transduction recognizes and transduces responses through phospholipid-derived molecules as second messengers, which are common to many classes of microbes, including beneficial and pathogenic microbes [6,7]. This defense response is initiated by the recognition of microbes, pathogen-associated molecular patterns (PAMPs), or danger-associated molecular patterns (DAMPs), consisting of endogenous elicitor molecules [5,8,9]. The immunity conferred by PAMPs or DAMPs is termed pattern-triggered immunity (PTI), and the immunity conferred after it is activated by the plant resistance response is termed effector-triggered immunity (ETI), which can converge in a similar innate immune response [6].

Today, growing evidence suggests that within the plant–microorganism interactions, many actors, including receptors, such as leucine-rich repeats (NLRs), lipid-transfer proteins, and phospholipid signaling networks, play relevant roles in detecting, transmitting, and responding to a given environmental threat by inducing properly dosed responses [10,11]. The activation of transcriptional dynamics [12,13,14], which includes small RNAs, lncRNAs and miRNAs [15,16,17,18,19], alternative splicing [20], changes to the phosphoproteome [21], protein stability-mediated SUMOylation or ubiquitination [22], hormones as mediators [23,24], DNA-regulated methylation [25], and phosphoinositide pools as signals [26], appears to be a common phenomenon during susceptible interactions; all of these factors may control variations between susceptible and resistant phenotypes differently when faced with various plant pathogens [27]. Changes in lipids during signaling could involve the biogenesis of fatty acids, sterols, glycerolipids [28], sphingolipids, and phospholipids [29,30,31]. The changes in lipids are usually related to responses to environmental conditions, such as temperature, salinity, and water disposition, and events related to pathogen effects [32,33]. All of this evidence points to a relevant role for phospholipases (PLs) in signaling pathways leading to disease resistance or the innate immune response.

Previous research in our group highlights the important role of phospholipase C (PLC)/diacylglycerol kinase (DGK) in the signal transduction pathway employed in cell suspensions. These roles include reducing phosphatidic acid (PA) levels (by almost 30%), PLC inhibition [34], and rapid activation of PLC transcription in mere minutes [35], all of which are responses to aluminum stress. Additionally, high levels of diacylglycerol pyrophosphate (DGPP) and PA were found in response to consortium infection involving PLC/DGK pathways [36]. This evidence establishes that PLDs produce another PA pool in response to different stresses. All of these findings point towards relevant crosstalk within PLC/PLD signaling pathways that lead to disease resistance or innate immune responses.

## 2. Actions of Phospholipases in Plants

### 2.1. Phospholipase C

PLC hydrolyses the phosphodiester bond on the glycerol side of phospholipids to produce diacylglycerol (DAG) and a phosphorylated head group [37]. PLCs have been classified based on their substrate or cellular function as whole enzymes with specific substrates, such as phosphatidylinositol (4,5) bisphosphate (PI(4,5)P_2_), or phosphatidylinositol (4) phosphate (PI(4)P) [38]. DAG is subsequently phosphorylated by a diacylglycerol kinase (DGK) into phosphatidic acid (PA), and then, PA or diacylglycerol pyrophosphate (DGPP) is employed as a second messenger during signaling based on the surrounding responses [29].

### 2.2. PLC Structure

PLCs in most organisms have been classified into multiple subfamilies; mammalian PLCs consist of 13 members grouped into the isoforms PLCβ, γ, δ, ε, η, and ζ, while plants have just one kind of PLC called PLCζ, which lacks the pleckstrin homology (PH) domain [39]. PLCs have been reported in many different species, including Arabidopsis. Using BLAST and PLC sequences from Arabidopsis as a query in plant databases, such as PlantGDB, SOL Genomics Network, EnemblPlants, Miyakogusa, and RAP-DB, different PLCs from plants have been identified [40,41,42,43,44,45,46,47,48,49,50,51,52,53,54,55,56,57,58,59,60,61,62,63,64,65,66,67,68,69,70,71,72,73,74]. Multiple PLCs have been registered in Arabidopsis (AthPLC1-9; *Arabidopsis thaliana* L. Heynh; [40]), grapevine (VviPLC1-5; *Vitis vinifera* L.; [41]), peach (PpePLC1-4; *Prunus persica* L. Stokes; [42]), papaya (CpaPLC1-4; *Carica papaya* L.; [43]), cucumber (CsaPLC1-4; *Cucumis sativus* L.; [44]), cassava (MesPLC1-4; *Manihot esculenta* Crantz; [45]), robusta coffee (CcaPLC1-4; *Coffea canephora* L.; [46]), black cottonwood (PotPLC1-4; *Populus trichocarpa*; [47]) and an external group, the moss Physcomitrella (PpPLC1-7; *Physcomitrella patens*; [48]). For some Solanaceae, 7 isoforms of PLC have been registered, including those in potato (StuPLC1-7; *Solanum tuberosum* L.; [49]), tomato (SolPLC1-7; *Solanum lycopersicum* L.; [50]), eggplant (SmePLC7; *Solanum melongena* L.; [51]), pepper (CanPLC1-7; *Capsicum annum* L. cv CM334; [52]), benthi (NbePLC1-7; *Nicotiana benthamiana* Domin; [53]), tobacco (NtaPLC1-7; *Nicotiana tabacum* L.; [54]) and petunia (PinPLC1-7; *Petunia integrifolia* Hook; [55]). For Gramineae, four isoforms of PLC have been registered in rice (OsaPLC1-4; *Oryza sativa* var. Japonica; [56]), stiff brome (BrdPLC1-4; *Brachypodium distachyon* (L.) P. Beauv; [57], great millet (SbiPLC1-4; *Sorghum bicolor* L. Moench; [58]), foxtail millet (SitPLC1-4; *Setaria italica* (L.) P. Beauv; [59]) and maize (ZmaPLC1-4; *Zea mays* L.; [60]). Finally, leguminous isoforms of PLC have been registered in soybean (GmaPLC1-3; *Glycine max* L. Merr; [61]) and wild legumes (LjaPLC1-4; *Lotus japonicus* L.; [62]).

PLCs from plants were subjected to phylogenetic analysis and organized into seven well-defined specific groups. PLC2 and PLC7 from many species formed a clade called type A (Figure 1). PLC1 and PLC3 formed another neighboring group called type B (Figure 1). PLC4 formed an exclusive group called type C, though PLC4 from Gramineae formed a separate group called type C´. PLC5 and PLC6 from Solanaceae formed a fourth clade called type D together with AthPLC6 (Figure 1). PLCs from monocots formed an exclusive group that included PLC1, PLC2 and PLC3 called type AB, and in the same manner, PLC3 from leguminous plants formed another exclusive group called type D’ (Figure 1). Finally, PLCs from the moss Physcomitrella formed a separate external group, as predicted. These findings indicated that highly conserved PLCs from different plants might have similar functions in the evolution of plants. Hence, the structures of PLCs from Solanaceae were different from those of Gramineae or leguminous PLCs, and these differences might be correlated with functional differences.

PLCs from plants presented a simple general structure with catalytic X and Y domains, an EF domain, and a C2 lipid-binding domain [63]. PLCs depend on the Ca^2+^ concentration in vitro (in a micromolar or millimolar range) [64], and are assumed to be in vivo substrates of PtdIns(4,5)P_2_ or PtdIns(4)P [33]. The I-TASSER suite was employed in a hierarchical approach to protein structure prediction and structure-based function annotation, and full-length atomic models of multiple PLCs were reconstructed by iterative template-based fragment assembly simulations (with server https://zhanglab.ccmb.med.umich.edu/I-TASSER/, accessed on 16 March 2021 [65]). PLCs from Arabidopsis and others showed the characteristic catalytic X and Y domain organization that forms a TIM barrel-like structure essential for their phosphoesterase activity [33]. AthPLC1 showed structural features that included an open TIM barrel-like (OBL; Figure 2) or a non-fist-like structure, while AthPLC3 featured an OBL structure modified without a lid (OBLm; Figure 2) or a non-fist-like structure without a thumb. AthPLC2 showed a closed TIM barrel-like (CBL; Figure 2) or a closed-fist structure, while AthPLC4 presented as a CBL structure with a handle (CB+H; Figure 2) or a closed-fist structure with a raised pinky, and AthPLC6 had a TIM barrel-like structure with or without a rod (CB-R; Figure 2) or a closed-fist structure without fingers. AthPLC7 showed a similar open TIM barrel-like structure as AthPLC1 and AthPLC5, while AthPLC8 and AthPLC9 were shown to have modified or incomplete TIM barrel-like structures with a handle (CB+Hm: Figure 2). In accordance with the phylogenetic arrangement, AthPLC1 and AthPLC3 shared an open TIM barrel-like structure to form clade type A (Figure 1). While AthPLC2 showed a closed TIM barrel-like structure, AthPLC7 did not; however, they shared a defined TIM barrel-like structure with small differences when grouped into clade type B (Figure 1). With respect to other PLCs, these PLCs showed changes in their TIM barrel-like structures or incomplete TIM barrel-like structures via modifications to the beta-sheet motif that resembled a bent thumb in AthPLC5 or an incomplete TIM barrel-like structure in AthPLC8 and AthPLC9; they all formed group C together with AthPLC4 (Figure 1). An exceptional case was AthPLC6, which resembles a rodless barrel and lacks internal beta sheet motifs inside the catalytic X and Y domains. With respect to other PLCs, these structures resembled PLCs from Arabidopsis, with an exception being PLC6 from Solanaceae, which presented an extra domain that resembled an external cap-like domain [40,66,67,68], and another strap-like domain similar to some Gramineae, such as great millet and foxtail millet. These protein clusters suggested different adaptative strategies in Solanaceae with respect to other plants, similar to monocots. Therefore, we hypothesized that different functions in transduction signaling based on specific PLC structures should be tested in the future.

With respect to other motifs, many proteins share a particular type of calcium-binding domain, known as the EF-hand. The EF-hand consists of an alpha-helical motif loop flanked on both sides by three other alpha-helical motifs, such as a spring domain. Some Glu or Asp residues are involved in ligand binding to Ca^2+^ and provoke a conformational change for activation or inactivation of the catalyzed enzymatic reactions for these enzymes [69]. The PLCs presented three types of EF-hands that resembled the thumb domain (Figure 2). While AthPLC1, 3, 4, and 7 showed thumb-up domains (TUs), AthPLC6 showed a bent-thumb domain (BT), and AthPLC2, 5, 8, and 9 showed no thumb domains (NT). Finally, the C2 lipid-binding domain was present in some PLCs from plants.

### 2.3. Nonspecific Phospholipase C

Whereas, PLCs use phosphoinositide specificity, nonspecific phospholipase C (NPC) uses common phospholipids, such as phosphatidyl-choline (PC) and phosphatidyl-ethanolamine (PE), as substrates to produce DAG and a corresponding phosphate-containing polar head group [70]. In addition to their inherent signaling functions, NPCs also play a role in lipid metabolism [33]. Based on sequence similarity with bacterial PC-PLC, six NPC genes (NPC1–NPC6) were identified in the Arabidopsis genome. NPCs possess a phosphoesterase domain necessary for esterase activity and three unknown domains that are highly conserved with a bacterial (*Mycobacterium tuberculosis*) PC–PLC [71]. The NPC backbone consists of a β-sheet surrounded by seven α-helixes and is in the non-membrane-spanning region. NPC participates in signaling, lipid metabolism, and development [72,73], and is involved in stress conditions, such as phosphate starvation [71], salt stress [72,74,75], aluminum toxicity [76,77,78], heat stress response [79], and infections with pathogens, such as *Pseudomonas syringae* [80].

### 2.4. Phospholipase D

PLD hydrolyzes phospholipids at the terminal phosphodiester bond, generating a free head group and PA. PLD mainly uses PE or PC, as well as others, as a substrate to produce PA, which alongside DGPP can be employed as a second messenger during signaling in plants [33]. PLDs participate in signaling, transport, and membrane degradation and are involved in stress conditions, such as drought stress (dehydration and salt stress), freezing, wounding, and pathogen interactions [81].

### 2.5. PLD Structure

PLDs have been organized into five subfamilies (α, β, γ, δ, ε and ζ), 12 of which have been described in Arabidopsis (Figure 3). PLD has two highly conserved domains called PLDc that are essential for its phosphoesterase activity [82]. While a characteristic C2 lipid-binding domain has previously been shown, C2-PLDs include PLDα, PLDβ, PLDγ, PLDδ, and PLDε and utilize PC, PE, and phosphatidylglycerol (PG) as substrates, but with different preferences [33]. PLDζ lacks the C2 domains and includes a pleckstrin homology (PH) domain near the N-terminus; it selectively uses PC as a substrate [83]. In plants, multiple PLDs have been found, such as in Arabidopsis (AthPLDα1-3, β1-2, γ1-3, δ, ε and ζ1-2; [83]) (Figure 3), grapevine (VviPLDα1-3(5), β(1), γ(1), δ1-3(3), ε(1) and ζ(1); [41]), peach (PpePLDα1-3(3), β(1), γ(1) δ1-2(2), ε(1) and ζ1-2(2); [42]), papaya (CpaPLDα1,3(2), β(1), γ1-1(2), δ1(2), ε(1) and ζ1-2(2); [43]), cucumber (CsaPLDα1-3(3), β(1), γ(1), δ1-2(2), ε(1) and ζ1-2(3); [44]), cassava (MesPLDα1-3(3), β(1), γ(1), δ1-3(7), ε(1) and ζ1-2(3); [45]), robusta coffee (CcaPLCDα1-3(4), β(1), γ(1), δ1-3(3), ε(1) and ζ1-2(2); [46]), black cottonwood (PotPLDα1-3(7), β1-2(2), γ(1), δ1-3(7), ε(1) and ζ1-2(3); [47]) and an external group, the moss Physcomitrella (PpPLDα1-2(5), β1-2(2), γ1-2(2), δ(1), ε(1) and ζ(2); [48]). For some Solanaceae, different isoforms have been registered in potato (StuPLDα1-3(6), β1-2(2), γ(1), δ1-3(5), ε(1) and ζ1-2(2), tomato (SolPLDα1-3(6), β1-2(2), γ(1), δ1-3(5), ε(1) and ζ1-2(2), eggplant (SmePLDα1-3(5), β1-2(2), γ(1), δ1-2(4), ε(1) and ζ1-2(2), pepper (CanPLCDα1-3(5), β1-2(2), γ1-2(2) δ1-3(4), ε(3) and ζ1-2(3), benthi (NbePLDα1-3(6), β1-2(4), γ1-2(2), δ1-3(8), ε(2) and ζ1-2(3), tobacco (NtaPLDα1-3(7), β1-2(4), γ1-2(2), δ1-3(9), ε(1) and ζ1-2(4) and petunia (PinPLDα1-3(5), β1-2(2), γ(1), δ1-3(5), ε(1) and ζ1-2(2). For Gramineae, some isoforms have been registered in rice (OsaPLDα1-3(7), β(1), γ(1), δ1-3(3), ε(1) and ζ1-2(3) [84,85,86,87,88,89,90,91,92,93,94], stiff brome (BrdPLDα1-3(4), β1(2), δ1-2(2), ε(1) and ζ1-2)(3), great millet (SbiPLDα1-3(6), β(1), γ(1), δ1-2(2), ε(1) and ζ1-2(4), foxtail millet (SitPLDα1-3(5), β1(1), γ(1), δ1-2(3), ε(1) and ζ1-2(3) and maize (ZmaPLDα1-3(5), β(1), δ1-2(3), ε(1) and ζ1-2(3) [95,96,97,98,99,100,101,102]. Finally, leguminous isoforms have been registered in soybean (GmaPLDα1-3(3), β1-2(3), γ(1), δ1(3), ε(2) and ζ1-2(3) [103,104,105], and wild legumes (LjaPLCDα1-3(6), β1-2(4), γ1-2(2), δ1-3(7), ε(3) and ζ1-2(4).

PLDs from plants contain two duplicated catalytic HKD (PLDc) motifs that interact with each other to form the active site, a C2 lipid-binding domain close to the N-terminus at approximately residue 130, which acts as a binding site for Ca^2+^ [83]. The Simple Modular Architecture Research Tool (SMART; http://smart.embl.de, accessed on 16 March 2021) was employed as a web resource for the identification and annotation of protein domains and the analysis of protein domain architectures [106]. All PLDs have the PLDc motif or HKD, but some isoforms have lost motifs (Figure 4). AthPLDα, β, γ, δ and ε showed C2, but not PH domains, while PLDζ showed PH, but not C2 domains [107,108,109,110,111,112,113,114,115,116,117,118,119,120,121,122,123,124,125,126]. In general, all PLDs from plants followed this trend, with the exception of extra PLDc2 and Phox homology (PX) domains in some PLDζs in Gramineae, such as stiff brome, great millet, foxtail millet, and maize, but not rice, and other types of different domains, such as RPTs, as internal repeats. When in Arabidopsis, all PLD structures were compiled using a swiss-model platform based on a crystal structure from AthPLDα1 called 6KZ9, as a template for the rest PLDα, β, γ, δ and ε, and finally, a structure of a catalytic domain from Human PLD named 6u8z and 6ohr were used as a base for PLDζ1 and 2, respectively (Figure 4). All PLDα, β, γ, δ and ε showed a similar structure to AthPLDα1, but with slight modifications, such as being without an internal alpha helical and without C2 domain structure in PLDζ1 and 2, respectively.

## 3. Phospholipid Signaling Responses Mediate the First Step during Immune Responses in Plants

Phospholipases have long been associated with responses to stress. As the first barrier against environmental agents, PLs crosstalk with PA and IP_3_ because they are involved in signaling events within many organisms [2,127,128,129]. Elicitors, such as peptidoglycans (PGNs) and lipopolysaccharide (LPS), from bacterial cell envelopes, bacterial elongation factor thermo-unstable (EF-TU), flg22 (a 22-amino acid peptide derived from bacterial flagellin), chitin from fungal cell walls, and glucans and glycoproteins in oomycetes have been linked to responses to biotic stress [130]. These elicitors can trigger physiological and morphological changes and generate reactive oxygen species (ROS) [10].

### 3.1. The Roles of PLCs in Immune Responses

Today, more evidence suggests that the PI-PLC family is required for HR-mediated defense responses by the induction of ETI- or PAMP-triggered immunity. Elicitor-induced PA accumulation was reduced by inhibitors of PLC activity, such as neomycin and U73122, and ROS production in tobacco cells was elicited [37,131]. A rice PLC isoform, *OsPI-PLC1*, was used to evaluate an incompatible interaction between a resistant genotype of rice and *Magnaporthe grisea*; its expression was only induced in BTH-treated rice seedlings or *Xanthomonas oryzae*-treated cell suspension cultures, suggesting that OsPLC1 plays important roles in signaling pathways leading to disease resistance in rice [132,133].

However, PLC isoforms from tomato, *SlPLC2* and *SlPLC5*, showed increased transcript levels upon inoculation with *Cladosporium fulvum* or when they were induced with fungal elicitors, such as xylanase or chitosan, suggesting general roles of *SlPLC2* and *SlPLC5* in the activation of plant defense responses with the involvement of nitric oxide (NO) in the regulation of these PLC genes and the subsequent defense response [27,134]. For SlPLC2, a role in the interaction with the necrotrophic fungus *Botrytis cinerea* was proposed, due to the increase in *SlPLC2* transcript levels together with those of *SlPLC3*, *SlPLC4*, and *SlPLC5* when specimens were inoculated with the necrotrophic fungus. PLC2 silencing via infection with tobacco rattle virus (TRV) also resulted in the reduced production of reactive oxygen species [135]. Previously, SlPLC2 was involved in the xylanase-induced expression of an SA-dependent PR-1a gene when SlPLC2-silenced plants were used, indicating that it plays a role in SA signaling [27,135]. All these data point to a nascent role of PLCs in biotic stress-related functions in which localized oxidative burst-dependent PA is required to infect the host.

Previous experiments demonstrated that SlPLC6 is important for resistance protein signaling following infections with *C. fulvum*, *Verticillium dahlia*, and *Pseudomonas syringae*, while *SlPLC4* is specifically and individually involved in the induction of a hypersensitive response (HR) triggered upon the detection of Avr4-carrying *Cladosporium fulvum* by the Cf-4 resistance protein [136]. In addition, the heterologous expression of SlPLC4 results in accelerated Avr4/Cf-4-induced HR in *N. benthamiana* [3]. For NbPLC2 from Nicotiana, increased susceptibility was registered when NbPLC2-silenced plants were challenged with a virulent strain of *Ralstonia solanacearum*. Plants confronted with infections with *R. solanacearum*, hrp-deficient *R. solanacearum*, *P. fluorescens*, or flg22 showed elevated NbPLC2 transcript levels [137]. In contrast, silencing NbPLC2s negatively affected the expression of PTI reporter genes, such as *NbPR-4* (a marker gene for jasmonic acid signaling), and decreased the levels of jasmonic acid and jasmonoyl-L-isoleucine after inoculation [137]. In addition, transcriptional dysregulation by several PTI inducers and effectors, oxidative burst, stomatal closure, and callose deposition were all reduced in the silenced plants. Silencing NbPLC2s negatively affected the expression of PTI reporter genes, which could mediate immune responses, leading to the suppression of bacterial infections [137].

### 3.2. The Action of PLDs in Cases of Biotic Stress

Current data indicate that PLDs produce the majority of pathogen-induced PA, and the individual contributions of different PLD isoforms to plant defense responses and their crosstalk with other signals are great challenges to resolve [138]. Some studies have reported decreased HR after recognition of elicitors via partial inhibition induced by n-butanol and several single-knockout PLD mutants. In particular, the PLDα, PLDβ, PLDγ and PLDδ isoforms displayed a related HR phenotype in the presence of n-butanol [139]. PLDα1-derived PA together with NADPH oxidase and heterotrimeric Gβ subunit (AGB1) proteins are required and function, in the same way, to fully resist different *P. syringae* pv. tomato DC3000 (avrRpm1) strains [140].

The establishment of the PLDβ1-deficient *P. syringae* pathosystem revealed lower levels of pathogen-induced PA production and increased levels of the lysophospholipids LPC, LPE, and LPG, which are involved in the pathogen- and wounding-induced responses [5]. PLDβ1-deficient plants exhibited increased resistance to PstDC3000, but increased susceptibility to *Botrytis cinerea* combined with an increase in PLA activity, suggesting crosstalk between the PLD and PLA pathways in plant-pathogen interactions [5]. Tomato *LePLDβ1* was induced by fungal elicitors, and RNAi knockdown of LePLDβ1 resulted in increases in the defense response and ROS production in PLDβ-suppressed tomato cells [81]. Knockdown of PLDß1 in rice increased resistance against *Pyricularia grisea* and *Xanthomonas oryzae* pv. oryzae cinerea, and the accumulation of reactive oxygen species in the absence of pathogen infection suggest that PLDß1 is a negative regulator of the immune response [141]. The pldγ1 and pldγ3 mutants had significantly elevated cell death responses following AvrRpm1 recognition [139].

Finally, PLDδ isoforms were more resistant against the penetration of nonadapted powdery mildew fungi (*Blumeria graminis* f. sp. Hordei, Bgh), while n-butanol-mediated inhibition of PA production by PLD action increased the penetration rate of Bgh spores on leaves [142]. The mode of action was discovered with PLDδ knockout plants, where they caused losses of ETI-induced and cell wall-based defense against Pst DC3000 (AvrRpm1) or a loss of MTI-induced cell wall-based defense against the nonhost powdery mildew *Erysiphe pisi*, suggesting the roles of PLD in plant–microbe interactions and defense responses [139]. Finally, a multitude of PLD isoforms in the HR triggered by elicitors are involved in triggering cell wall-based defenses against pathogen infection and suggest that different PLDs present in common physiological processes have different modes of action.

### 3.3. The Action of PLCs in Viral Replication

During the replication of red clover necrotic mosaic virus (RCNMV) in *Nicotiana benthamiana*, NbPLDα, and NbPLDβ and their derived PA were required for viral RNA replication. Consistent with these data, exogenous application of PA enhanced viral RNA replication in plant cells and plant-derived cell-free extracts [143]. Finally, curated PlaD data were analyzed, and the global landscape of public pathogenesis-related genes from the model plant Arabidopsis and three major crops (maize, rice, and wheat) was explored. Here, only PLAs, not PLDs or PLCs, were induced by viral treatment [68].

## 4. Phospholipid Signaling Response as a Means to Alleviate Other Stresses

Phospholipid signaling can contribute to the regulation of other stresses, potentially playing an important role in responses to abiotic stress. For PI-PLC isoforms, all AtPLCs, with the exception of AtPLC8, showed increased or decreased transcriptional expression in some microarrays in response to various environmental stimuli, such as cold, drought, salinity, water deficit or dehydration, heat stress, or thermotolerance [40,41,42,43,44,45,46,47,48,49,50,51,52,53,54,55,56,57,58,59,60,61,62,63,64,65,66,67,68,69,70,71,72,73,74].

### 4.1. Roles of PLC in Temperature Changes

An example of the role of phospholipids, such as phosphatidylinositol in the binding of signaling-related proteins, was observed through phosphorylation assays after 15 min of cold exposure, suggesting that phospholipids are part of the very early response after temperature drop [144]. One of the molecules considered a signaling molecule regulated by cold is PA, a product of DAG and the hydrolysis of PI(4,5)P_2_ by enzymes, such as PLC2. With respect to temperature changes, AtPLC3 and AtPLC9 were shown to play roles in thermotolerance by increasing IP_3_ levels with changes to intracellular Ca^2+^ and presented an additive effect on regulating heat stress [145,146,147].

### 4.2. Roles of PLC in Osmotic Stress

Plants use different strategies to cope with osmotic stress, and lipid signaling has been implicated as one of the factors in various plant systems. PLC3 overexpression conferred tolerance to drought stress accompanied by decreased sensitivity to ABA-induced stomatal closure in *Arabidopsis thaliana* and other plants, such as maize and tobacco [148]. PLC4 in *Arabidopsis thaliana* has been described as a negative regulator of signaling, such as salt-induced Ca^2+^ signaling [149]; similarly, the action of PLC9 is implicated in changes in IP_3_-mediated regulation of heat tolerance and Ca^2+^ requirements [145]. In rice, OsPLC1 elicited Ca^2+^ signals regulating salt tolerance, as OsPLC1 was translocated from the cytosol to the plasma membrane, where it can hydrolyze PtdIns4P [150]. Overexpression of PI-PLC2 from *Brassica napus* into canola induced significant changes in the expression profiles of stress-related genes and enhanced drought tolerance [151]. Additionally, overexpression of ZmPI-PLC1 enhanced the grain yield of maize under drought conditions, while suppression of ZmPI-PLC1 had the opposite effect [152].

### 4.3. Action of PLD in Osmotic and Drought Stresses

In plants, osmotic stress-triggered stomatal closure requires PLD action and crosstalk with third-messenger gaseous signaling molecules, such as hydrogen sulfide (H_2_S) in *Arabidopsis thaliana*. PLDα1 and H2S alleviate osmotic stress by suppressing ROS and maintaining membrane integrity [153]. When PLDδ was associated with a wider stomatal aperture under osmotic stress, a lower H2S content or expressed L-cysteine desulfhydrase was shown (LCD; [154]). In both cases, PLD action enhances the alleviation of osmotic stress by triggering stomatal closure, which could reduce transpiration, prevent water loss, and maintain cell turgor pressure under drought stress [153,154]. Li and colleagues reported lipid profiling results and revealed that PLDδ contributed approximately 20% of the phosphatidic acid produced in plants during freezing and that overexpression of PLDδ increased the production of phosphatidic acid species [144]. Moreover, the knockdown of PLDα1 was related to a decrease in the level of PA during freezing-induced hydrolysis of phosphatidylcholine, whereas overexpression increased freezing tolerance [155]. PLDα1 deficiency rendered plants insensitive to ABA and induced stomatal closure [156]. Knockout of PLDα3 rendered Arabidopsis more sensitive to salt stress, while overexpression of PLDα3 enhanced salt tolerance. It has been proposed that PLDα3 positively mediates plant responses to hyperosmotic stresses in a negative manner in response to ABA [157]. Currently, there is another interesting interaction in plants that is mediated by PLs action and nitric oxide (NO). NO enhances tolerance to abiotic stress by increasing proton pump and/or antiport activities and reducing hydrogen peroxide (H_2_O_2_) levels; a PLD inhibitor (1-butanol) diminishes these effects. This suggests an important role of PLD signal transduction in this process [158,159,160]. 

## 5. Conclusions and Perspectives

In this review, information on phospholipases was compiled for a wide range of plant species. Analyses of the phylogenetic relationships, protein structures, and domains of PLCs indicated that PI-PLCs were highly conserved compared with their homologous genes. Additionally, other PC-PLCs (NPCs) and PLDs were analyzed, which showed that PLs may have important functions in signal transduction during biotic and abiotic stresses. Taken together, these results provide useful information for further study of the roles and functions of PLs in plants under environmental stress conditions, including innate immune responses.

## Figures and Tables

**Figure 1 plants-10-00921-f001:**
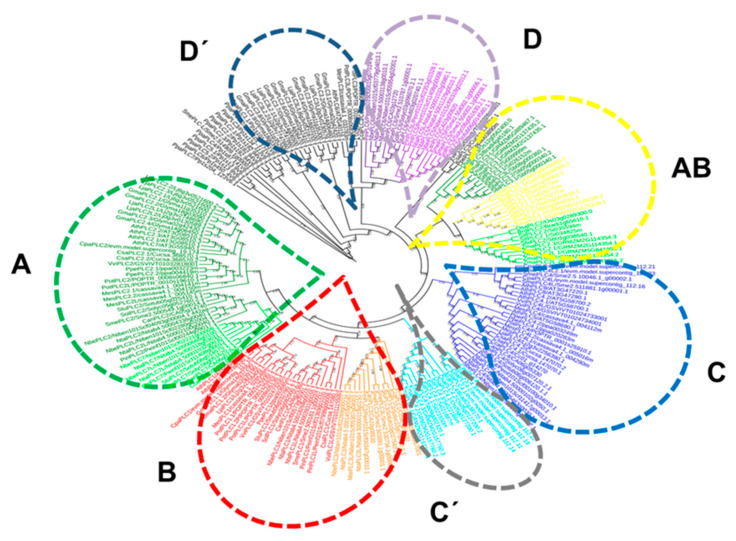
Phylogenetic tree of PLCs from plants. The phylogeny was reconstructed based on an alignment and was made by using the maximum parsimony method, with 1000 bootstrap replicates; the results were visualized using MEGA X and ITOL. Group A comprises PLC2 and PLC7; group B comprises PLC1 and PLC3; group AB comprises PLC1, PLC2, and PLC3 from Gramineae; group C comprises PLC4; group D comprises PLC6 and PLC5 from Solanaceae together with AthPLC6; an exclusive group C’ comprises PLC4 from Gramineae; and group D’ comprises PLC3 from legumes. Finally, the moss Physcomitrella forms an external group. The numbers at the nodes are the bootstrap values (>50%), and the branch lengths from the root are displayed.

**Figure 2 plants-10-00921-f002:**
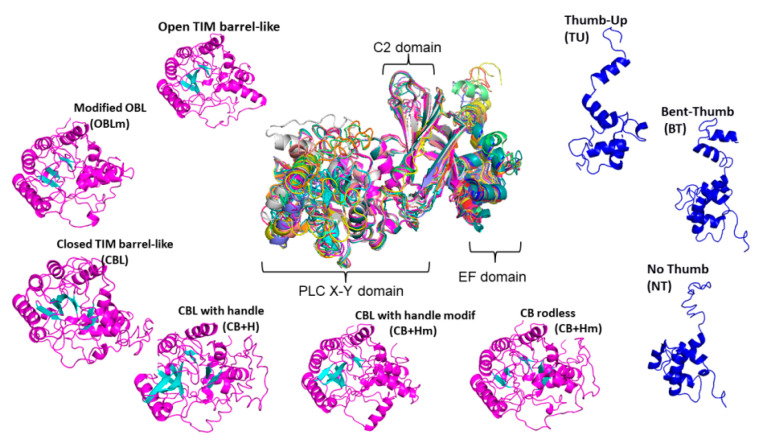
Postulated TIM barrel-like PLC structural types. Predicted PLC structures were classified as open TIM barrel-like (OBL); modified OBL (OBLm); closed TIM barrel-like (CBL or CBLm); CBL with handle (CB+H or CB+Hm); and TIM barrel-like without an internal beta sheet motif (CB-R). In accordance with modifications in the EF-hand domains, three kinds of structures were predicted: Thumb-up domain (TU), bent-thumb (BT), or no thumb domain (NT).

**Figure 3 plants-10-00921-f003:**
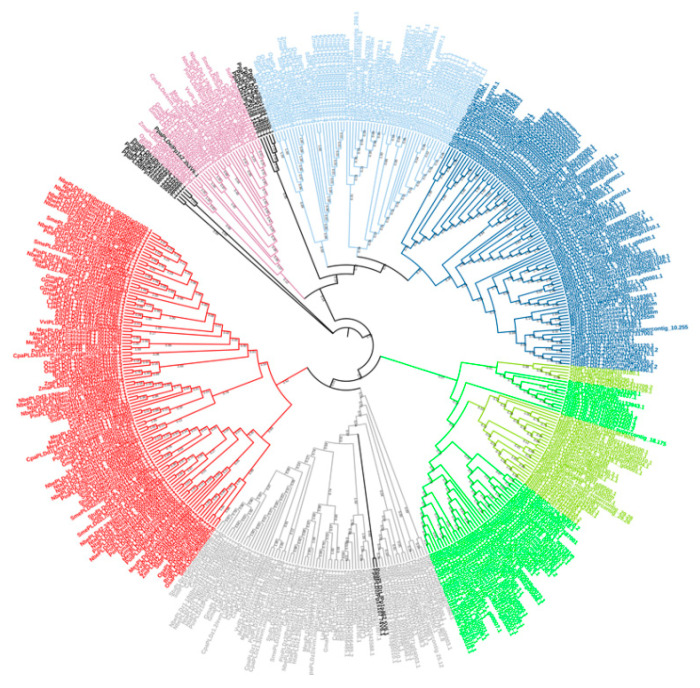
Phylogenetic tree of PLDs from plants. The phylogeny was reconstructed based on an alignment and was made using the maximum parsimony method, testing with 1000 bootstrap replicates and displayed using MEGA X and ITOL tools. They were shown as follows, for PLDα in blue [1 and 2 (dark ones) and 3 (light one)], PLDβ in dark green, PLDγ in light green, PLDδ in red, PLDε in purple, and PLDζ in gray. Finally, in an external group, the moss *Physcomitrella*; the numbers at the nodes are the bootstrap values (>50%), and the branch lengths from the root were displayed.

**Figure 4 plants-10-00921-f004:**
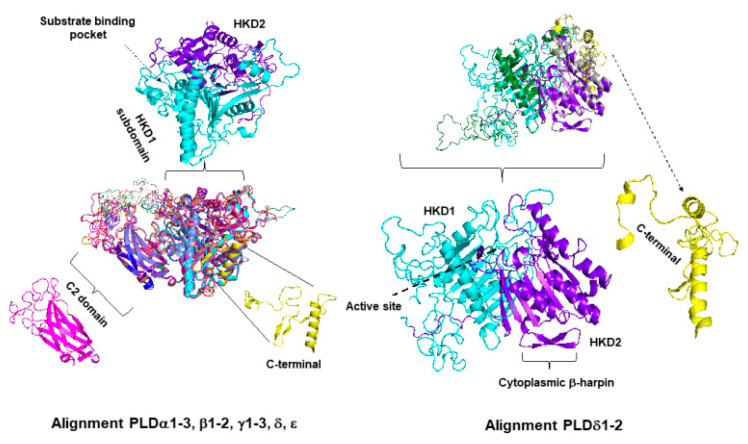
Postulated PLD structure types. Predicted PLD structures were compiled in a swiss-model platform based on a crystal structure from *Arabidopsis* phospholipase D alpha called 6KZ9 as a template for PLDα, β, γ, δ and ε and structure of a catalytic domain from Human PLD named 6u8z and 6ohr for PLDζ1 and 2, respectively.

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
