# Peer review of "Phospholipases C and D and Their Role in Biotic and Abiotic Stresses"

_plants, 2021, doi:10.3390/plants10050921_

Round 1

Reviewer 1 Report

very good review but I missed something like the group's contribution to the subject. 

A paragraph about their understand could make it better.

Author Response

A paragraph including the contribution of our group has been included, lines page 2, line 66.

Reviewer 2 Report

In the current work entitled "Phospholipases and their role in the innate immune response", by Gonzalez-Mentoza et al., the authors provided recent scientific data regarding this topic.
In general, the script should be checked by a native English speaker, in order to facilitate the reader.
The manuscript presents several key aspects regarding the crucial role of Phosholipases in plant adaptability to several exogenous stimuli. 
The authors should include more information regarding section 4. "Phospholipid signaling response as a means to alleviate other stresses". Especially section 4.3. The authors should also check the role of Nitric Oxide (NO) since this biomolecule regulates plasma membrane H-ATPase activity in several plants during drought stress via the induction of PLD-derived phosphatidic acid (PA) formation. Add more data concerning the role of NO.
Also, more scientific data should be added related to the interplay of PCD and Proline biosynthesis under osmotic stress syndromes.
Also, in the same section (4.3) the authors should be more specific about the "plants". Name the plants in the same way that it is done in section 4.2.
Overall the current work is worthy of being published after the proposed corrections.

Author Response

More information regarding section 4.3 has been included in page 11 line 434 and the name of the plants were named in the same way along the manuscript.

Reviewer 3 Report

The manuscript by González-Mendoza et al. reviews the function of phospholipase C and D in biotic and abiotic stresses.

Comments:

  1. The title of this review manuscript is “Phospholipases and their role in the innate immune response. However, the authors mainly talk about the function of phospholipase C and D. In addition to the innate immune response, the roles of PLC and PLD in environmental stresses are also discussed here. I suggest adding more content about PLA or change the title to “Phospholipase C and D and their role in biotic and abiotic stresses”.
  2. Similar to figures 1 and 2, a phylogenetic tree of PLDs and PLD structure will be helpful here.

Author Response

The title has been changed according to the reviewer suggestions to: Phospholipases C and D and their role in biotic and abiotic stresses.

Similar figures to PLC figures 1 and 2, were included for PLD, figures 3 and 4, page 7, line 256, and page 8, line 287.  

The manuscript has been reviewed by AJE in order to improve the English of the text.